# Chemotaxonomic Variation in Volatile Component Contents in Ancient *Platycladus orientalis* Leaves with Different Tree Ages in Huangdi Mausoleum

**DOI:** 10.3390/molecules28052043

**Published:** 2023-02-22

**Authors:** Bei Cui, Tao Zheng, Ping Deng, Sheng Zhang, Zhong Zhao

**Affiliations:** 1College of Forestry, Northwest Agriculture and Forestry University, Xianyang 712100, China; 2Research Center for the Conservation and Breeding Engineering of Ancient Trees, Xianyang 712100, China; 3College of Biology Science and Engineering, Shaanxi University of Technology, Hanzhong 723001, China; 4College of Biology and Pharmacy, Yulin Normal University, Yulin 537000, China

**Keywords:** ancient *Platycladus orientalis*, HS–SPME–GC–MS, essential oil, volatile component, tree age

## Abstract

To gain insight into the differences in the composition and volatile components content in ancient *Platycladus orientalis* leaves with different tree ages in Huangdi Mausoleum, the volatile components were identified by headspace solid-phase microextraction combined with gas chromatography-mass spectrometry (HS–SPME–GC–MS) method. The volatile components were statistically analyzed by orthogonal partial least squares discriminant analysis and hierarchical cluster analysis, and the characteristic volatile components were screened. The results exhibited that a total of 72 volatile components were isolated and identified in 19 ancient *Platycladus orientalis* leaves with different tree ages, and 14 common volatile components were screened. Among them, the contents of α-pinene (6.40–16.76%), sabinene (1.11–7.29%), 3-carene (1.14–15.12%), terpinolene (2.17–4.95%), caryophyllene (8.04–13.53%), α-caryophyllene (7.34–14.41%), germacrene D (5.27–12.13%), (+)-Cedrol (2.34–11.30%) and α-terpinyl acetate (1.29–25.68%) were relatively higher (>1%), accounting for 83.40–87.61% of the total volatile components. Nineteen ancient *Platycladus orientalis* trees were clustered into three groups through the HCA method based on the 14 common volatile components content. Combined with the results of OPLS–DA analysis, (+)-cedrol, germacrene D, α-caryophyllene, α-terpinyl acetate, caryophyllene, β-myrcene, β-elemene and epiglobulol were the differential volatile components to distinguish ancient *Platycladus orientalis* with different tree ages. The results revealed that the composition of the volatile components in ancient *Platycladus orientalis* leaves with different tree ages was different, showing different aroma characteristics, which provided a theoretical reference for the differential development and application of volatile components in ancient *Platycladus orientalis* leaves.

## 1. Introduction 

*Platycladus orientalis* (L.), belonging to Cupressaceae, known as *Chamaecyparis* and *Thuja occidentalis* L., is a unique evergreen coniferous tree in China [1,2]. It is also an important landscaping tree species, widely distributed in various regions of the country [3,4]. Ancient *Platycladus orientalis* tree resources are very abundant, and there are hundreds of years or even thousands of years old trees in many areas, most of which have developed into tourist attractions due to their unique cultural value [5,6]. The *Platycladus orientalis* groups of Huangdi Mausoleum in Yan’an City, Shaanxi Province, were the oldest, largest and most completely preserved pure forest plantation and ancient tree population in China [7]. The tree age of the oldest existing, ‘the *Platycladus orientalis* tree planted by Huangdi,’ was 5000 years, which was known as the ‘father of the world Cypress.’

*Platycladus orientalis* grew slowly, but *Platycladus orientalis* wood was durable, and its color, aroma and antibacterial properties were valuable [8,9]. Most *Platycladus orientalis* contained essential oils with a specific aroma, and some essential oils extracted from leaves, wood or berries were used as the main aromatics in many consumer products [10,11]. *Platycladus orientalis* leaves contain a variety of medicinally active ingredients, including flavonoids, volatile oil, and tannins, with an abundance of pharmacological effects such as antibacterial, anti-tumor, hemostasis, hair growth promotion and anti-inflammatory [12,13]. The volatile oil extracted from the *Platycladus orientalis* leaves was widely used in industry as a flavoring agent in woody, spicy and other flavors and other products and had the effects of detoxification, soothing the nerves, analgesia and antibacterial [14,15]. It had broad development prospects and practical significance to fully explore the medicinal value of *Platycladus orientalis* leaf products, carry out deeper research and development, and develop high-value-added *Platycladus orientalis* leaf active ingredient products [16,17]. Lin et al. (2015) used different essential oil extraction methods to extract volatile oils from the leaves of five species of Cupressaceae plants [18]. Comparing the contents of volatile oils extracted by different extraction methods, it was found that the highest yield of volatile oil was obtained by steam distillation [19]. The volatile oils components were analyzed and identified by GC–MS, and it was reported that most of the compounds in the volatile oils of Cupressaceae plants were terpenes, and the contents and components of volatile oils of different Cupressaceae plants were quite different [20,21]. Kim et al. (2015) also used steam distillation to extract volatile oil from the leaves of five different varieties of *Juniperus chinensis*, and 21 volatile components were identified by GC–MS analysis [22]. It was found that the essential oil of *Sabina* was mainly composed of monoterpenes, including bornyl acetate and sabinene. The main component of *Platycladus orientalis* essential oil was cedrol, which had an activation effect and could be used as an antagonist of platelet-activating factor (PAF) receptor, and played an important role in the inflammatory response, human respiratory system and cardiovascular disease [16,23].

The active components of volatile plant oil were affected by region, climate and environment [24,25]. Scholars at home and abroad have found that there are obvious regional differences in volatile components in *Platycladus orientalis* leaves from different habitats [20,26]. However, there are few studies on the influence of tree age on the composition of volatile components in ancient *Platycladus orientalis* leaves. Headspace solid phase microextraction and gas chromatography-mass spectrometry (HS–SPME–GC–O–MS) is a method for the detection of volatile or semi-volatile components, odorous substances and aroma substances [27,28,29]. Therefore, the essential oil of ancient *Platycladus orientalis* leaves of different ages was extracted by steam distillation, and its volatile chemical components were analyzed by the HS–SPME–GC–O–MS method. The active components of essential oil of ancient *Platycladus orientalis* leaves of different ages were compared, which was of great significance for the development of high-value-added products of essential oil of ancient *Platycladus orientalis* leaves.

In the present study, the ancient *Platycladus orientalis* leaves of different ages in Huangdi Mausoleum were used as research materials. The essential oil content of ancient *Platycladus orientalis* leaves was extracted by steam distillation, and the composition of volatile components in leaves was analyzed by HS–SPME–GC–MS. Orthogonal partial least squares, principal component analysis and hierarchical cluster analysis were used to reveal the differences in volatile components in ancient *Platycladus orientalis* leaves of different ages. The results provided a reference for the differential development and utilization of essential oil in ancient *Platycladus orientalis* with different tree ages.

## 2. Results

### 2.1. Essential Oil Yield of Ancient Platycladus orientalis Leaves with Different Tree Ages

The essential oil extracted by steam distillation was pale yellow, and the yield of essential oil increased first and then decreased with the increase of tree age. The highest yield of essential oil was tree age with 3000 years old (1.810%), followed by 2500 years old (1.463%). Through the sensory comparison of essential oil in *Platycladus orientalis* leaves with different tree ages, the essential oil of *Platycladus orientalis* leaves with the tree age of 2500–3000 years was darker and had a more lasting mellow, rich wood fragrance and no irritating head fragrance and odor. However, the essential oils of *Platycladus orientalis* leaves with a tree age of 2000 and 5000 years were lighter in color; although they had an aromatic flavor, it was not lasting, and they even had other odors. A significant difference in the extraction yield of essential oil of ancient *Platycladus orientalis* leaves with different tree ages (*p* < 0.05) was observed (Figure 1), which might be derived from the physiological metabolic capacity of ancient *Platycladus orientalis* with different tree ages. In order to further understand the reasons for the significant differences in essential oils, the HS–SPME–GC–MS method was applied to qualitatively and quantitatively analyze the volatile components in ancient *Platycladus orientalis* leaves with different tree ages.

### 2.2. Analysis of Volatile Components in Ancient Platycladus orientalis Leaves with Different Tree Ages

The volatile components in ancient *Platycladus orientalis* leaves with different ages were determined by gas chromatography-mass spectrometry, and a total of 82 compounds were detected (Appendix A), with the most terpenes compounds (53, Appendix A), followed by alcohols (19, Appendix A), esters (6, Appendix A), aldehydes (3), ketones (1). Significant differences in the quantity and composition of volatile components in 19 ancient cypress leaves were observed. The volatile components detected were further classified, of which the terpene compounds were rich in content and variety and were the main component, accounting for more than 66.49% of the total volatile components. Alcohols and esters also account for a large proportion of volatile components. There were significant differences in the volatile components of ancient *Platycladus orientalis* leaves of different tree ages, which were manifested as changes in the content of components, and a few components only existed in leaves of specific tree ages. For example, longifolene- (V4) was a specific component of HDL 1, and nerol only exists in HDL 2, HDL 6 and HDL 12, D-verbenone only exists in HDL 6 and HDL 18, and andrographolide exists in HDL 5, HDL 7 and HDL 16.

As shown in Figure 2, the overall profile of each volatile component was described in the heat map through a color box. The color intensity was based on a standardized scale from a minimum of 0.0 (red) to a maximum of 25.0 (blue), which indicated that the abundance of the volatiles was from low to high. The HCA results exhibited that the 19 ancient *Platycladus orientalis* trees with different tree ages were obviously divided into two groups, among which 5000 years and 3000 years were divided into one group, 2000 years and 2500 years were clustered into another group, which was consistent with the sensory results. In addition, based on the tree diagram, the volatile components could be divided into three categories. Group Ⅰ included six compounds with high content in 19 samples, including α-pinene, caryophyllene, α-caryophyllene, germacrene, 3-carene and α-terpinyl acetate, with the taste of wood, flowers and citrus. Group Ⅱ was composed of 11 higher-content compounds, including terpinolene, (+)-cedrol, sabinene and β-myrcene, with herbaceous, sweet and spicy flavors. The remaining 65 compounds were classified as group Ⅲ, with low content in ancient *Platycladus orientalis* leaves.

The volatile components in ancient *Platycladus orientalis* leaves with the tree age of 2000 a were mainly small molecular aromatic substances, such as caryophyllene, α-caryophyllene, germacrene D, and the volatile components in ancient *Platycladus orientalis* leaves with the tree age of 2500 a are mainly 3-carene and terpinolene. Whereas, with the tree age of 3000 to 5000 a, the volatile components were mainly sabinene, (+)-cedrol, and α-terpinyl acetate. With the increase of tree age, the aromatic substances in ancient *Platycladus orientalis* leaves were not only enriched but also transformed into a more stable chemical state, from small molecular terpenes to alcohols and esters with longer-lasting fragrances and larger molecular weight. The relative content of α-pinene, 3-carene, terpinolene and β-myrcene increased first and then decreased with the increase of tree age. With the increase of tree age, the relative content of caryophyllene and α-caryophyllene decreased gradually. The relative content of macromolecular aromatic substances such as (+)-cedrol and α-terpinyl acetate increased gradually with the increase of tree age. It might be inferred that in the sapling stage, the tree vigor is strong, and the nutrient supply is biased towards vegetative growth. With the increase of tree age, the tree vigor tends to be stable, the nutrient distribution of the tree tends to be balanced, the metabolism ability of branches and leaves is strong, and the flavor becomes stronger. It could be seen that under the same conditions of quality, origin and preparation method, the composition and relative content of volatile components in ancient *Platycladus orientalis* leaves at different tree ages were different.

A total of 14 common components were screened from 19 ancient *Platycladus orientalis* leaves, including α-pinene, sabinene, 3-carene, terpinolene, caryophyllene, α-caryophyllene, germacrene D, (+)-cedrol, α-terpinyl acetate, (−)-terpinen-4-ol, β-thujene, β-pinene, β-myrcene and caryophyllene oxide (Table 1). Among them, nine main components were higher than 1% of the total volatile components content. They were α-pinene (6.88–16.76%), sabinene (1.11–7.29%), 3-carene (1.14–15.12%), terpinolene (2.31–4.95%), caryophyllene (8.04–13.53%), α-caryophyllene (7.54–14.41%), germacrene D (5.57–13.13%), (+)-cedrol (2.34–6.89%), α-terpinyl acetate (1.29–25.68%), β-myrcene (1.44–2.52%), of which these components accounted for more than 67.89% of the total volatile components.

The variation coefficients of volatile components in ancient *Platycladus orientalis* leaves at different ages demonstrated that the variation coefficients of 14 volatile components ranged from 12.72% to 70.77%, with an average variation coefficient of 33.75%. The variation coefficients of α-terpinyl acetate were the largest, followed by caryophyllene oxide, indicating that α-terpinyl acetate and caryophyllene oxide were significantly different in leaves at different tree ages. The variation coefficients of terpinolene, caryophyllene, α-caryophyllene and β-myrcene were the smallest, indicating that their contents were relatively stable.

### 2.3. HCA and PCA

Principal component analysis (PCA) was applied to use several principal components to reveal the accumulation differences and variability of volatile components in ancient *Platycladus orientalis* leaves with different tree ages. PCA was performed on the samples to preliminarily understand the overall volatile substances between the volatile components in ancient *Platycladus orientalis* leaves with different tree ages and the variability between the samples in the group. In this study, 4 principal components (PC1, PC2, PC3 and PC4) were extracted, which were 49.99%, 28.35%, 8.33%, and 5.38%, respectively, and the cumulative contribution rate reached 92.05% (Figure 3A). Nineteen ancient *Platycladus orientalis* trees were divided into three groups. In the PCA-2D diagram, the clustering of samples could be seen more intuitively. Based on PCA results, it was demonstrated that the difference in volatile component contents between samples might be the difference between *Platycladus orientalis* leaves samples with different tree ages. Caryophyllene, α-caryophyllene, germacrene D, (+)-cedrol and epiglobulol contributed more to PC1, while sabinene, caryophyllene oxide, β-elemene and α-terpinyl acetate contributed more to PC2.

The content of essential oil and the relative content of 14 identified common volatile components in 19 ancient *Platycladus orientalis* leaves were normalized by unit variance scaling (UV Scaling). The word clustering method was performed to calculate the distance (similarity) between different categories of data points by using Euclidean distance, and the clustering tree diagram of 19 ancient *Platycladus orientalis* was constructed. As could be seen from Figure 3B, when the distance was five, it could be clustered into three groups. The first group contained HDL1 (5000 a), and the contents of (+)-cedrol, epiglobulol and caryophyllene oxide in HDL1 were the highest. The second group consisted of six ancient *Platycladus orientalis* with tree ages of 3000 years (HDL2, HDL4, HDL10, HDL11, HDL14, and HDL15). Among them, sabinene and α-terpinyl acetate contents were higher. The third group contained six ancient *Platycladus orientalis* samples with tree ages of 2500 years (HDL3, HDL6, HDL7, HDL9, HDL13, and HDL18) and six samples with tree ages of 2000 years (HDL5, HDL8, HDL12, HDL16, HDL17, and HDL19), of which caryophyllene, α-caryophyllene and germacrene D were higher.

### 2.4. OPLS–DA Analysis

Orthogonal partial least squares discriminant analysis (OPLS–DA) is a multivariate statistical analysis method for supervised pattern recognition, which combines orthogonal signal correction (OSC) and PLS–DA methods, and can decompose the X matrix information into two types of information related to Y and irrelevant, and screen the different variables by removing irrelevant differences [30,31]. OPLS–DA analysis was carried out to further investigate the difference in volatile components in ancient *Platycladus orientalis* leaves with different tree ages. 

The 19 samples in the experiment were pre-grouped based on different tree ages, and the OPLS–DA mathematical model was constructed by R software. It could be seen from Figure 4A that the cumulative variance contribution rate of the first two principal components of OPLS–DA was 62.8%, which integrated most of the information of the volatile components in 19 ancient *Platycladus orientalis* leaves with different tree ages and could better explain the variation trend of volatile substances. Based on the scores of volatile components in ancient *Platycladus orientalis* leaves with different tree ages on the first and second principal components, the leaves samples of four different ages were completely separated on the first principal component (46%). The permutation test is mainly used to verify the fitting degree of the OPLS–DA model. R^2^Y and Q^2^ are applied to verify the interpretation rate and prediction ability of the model. The closer the values of R^2^Y and Q^2^ are to 1, the better the interpretation rate and prediction ability of the model are. A total of 200 random permutation and combination experiments were carried out on the established model. The Q^2^ and R^2^ Y of the OPLS–DA model reached 0.944 and 0.961, respectively, indicating that the established model had a strong discrimination function and prediction ability (Figure 4B). In addition, the results of 200 random permutation experiments displayed that the left points of R^2^Y and Q^2^ were lower than the right points, and R^2^Y > Q^2^, indicating that there was no over-fitting phenomenon in the established model, which could be used to screen the key differential substances of volatile components in ancient *Platycladus orientalis* leaves with different tree ages.

### 2.5. Analysis of Differential Volatile Components in 19 Ancient Platycladus orientalis Leaves with Different Tree Ages

Combined with the loading plot of OPLS–DA and VIP values (variable important for the projection) analysis, the differential markers of volatile components in 19 ancient *Platycladus orientalis* leaves with different tree ages could be identified. The VIP values could quantify the contribution of each variable factor to the classification. The larger the VIP value, the more significant the contribution of volatile components to the difference between ancient *Platycladus orientalis* leaves samples with different tree ages. Though calculating the VIP values, it was found that the VIP values of eight common volatile components were greater than 1 (Figure 4C). Cluster analysis of 19 ancient *Platycladus orientalis* with different tree ages was carried out with eight selected key volatile components, and a heat map was described. It could be seen from Figure 4D that the 19 ancient *Platycladus orientalis* with different tree ages were obviously divided into three categories, and the relative content of key volatile components was significantly different (*p* < 0.05). (+)-Cedrol, epiglobulol and caryophyllene oxide were the characteristics of type I samples, α-Terpinyl acetate and Sabinene were the characteristics of type II samples, β-elemene, β-myrcene, germacrene D, α-caryophyllene and caryophyllene were the characteristics of type III samples, indicating that these eight key volatile components played an important role in distinguishing ancient *Platycladus orientalis* with different tree ages.

## 3. Discussion

The number of volatile components in ancient *Platycladus orientalis* leaves was significantly different among the four tree ages, and the volatile component contents varied with tree age. Through the determination of composition and content by GC–MS, the volatile components *in ancient Platycladus orientalis* leaves were mainly monoterpenes, sesquiterpenes and their derivatives. α-pinene, 3-carene, caryophyllene, α-caryophyllene, daurene, and terpinyl acetate were the main components in ancient *Platycladus orientalis* leaves, and their relative contents were as high as 72.45–88.93%. Precursors of sesquiterpenes and their derivatives, such as caryophyllene, germacrene D, cedar alcohol and eucalyptol, are synthesized in cytoplasm mainly through mevalonate (MVA) pathway [32,33]. Whereas the precursors of monoterpenes and their derivatives, such as α-pinene, β-myrcene, 3-carene, caryophyllene, and α-caryophyllene, were mainly synthesized in the plastids through the 5-phosphate deoxyxylulose/2C-methyl 4-phosphate-4D-erythritol (MEP) pathway, and then various terpene compounds are formed under the action of the corresponding terpene synthase, and the production of downstream terpene compounds is determined by the role of key enzymes [34,35]. The yield of essential oil from ancient *Platycladus orientalis* leaves was significantly affected by tree age. The precursors of essential oil components in leaves were mainly generated in plastids, which were then disintegrated by material transformation and then entered the cytoplasm [36]. The content of α-pinene, 3-carene, caryophyllene, α-caryophyllene, and myrcene decreased with the increase of tree age, and the number or activity of key enzymes decreased. The plastids in the cells were decomposed and further produced into various substances, such as cedrol and terpinyl acetate in plants, which may be caused by the mutual conversion of secondary metabolites in plants [37,38].

OPLS–DA and HCA are two important multivariate statistical analysis methods that can reduce the dimension of more variables to as few new variables as possible so that these new variables are not related to each other and screen out differential markers [39]. At present, they are widely used in quality difference analysis and genetic identification of food, medicine and agricultural products [40]. Liu et al. used PCA and PLS–DA analysis methods to analyze the physical and chemical components of different grades of sun-dried green tea, and it was found that the content of amino acid components in tea was an important physical and chemical component for the classification of sun-dried green tea, of which lysine (Lys), proline (Pro) and phenylalanine (Phe) played a more significant role [41]. Lota et al. (2001) reported that 15 kinds of mandarins were classified into three chemical types, namely limonene type, limonene/γ-terpinene type and linalyl acetate/limonene type through PCA and HCA analysis [42]. The difference in the macamide content of *Maca* from different origins was determined by liquid chromatography-ultraviolet detection tandem mass spectrometry, and the regions were distinguished based on the PLS-DA model [43]. The essential oils of Liangping pomelo peel at five different tree ages were clustered into three categories by HCA analysis. Based on the characteristic volatile substances, the three types of essential oils can be divided into plant oil flavor, typical citrus sweet flavor and rich mint flavor [44]. Combined the HCA results with the VIP values in the OPLS–DA, the load map suggested that the 19 ancient *Platycladus orientalis* leaves of different ages were distinguished the differential characteristic volatile components, including (+)-cedrol, germacrene D, α-caryophyllene, α-terpinyl acetate, caryophyllene, β-myrcene, β-elemene and epiglobulol, and 19 ancient *Platycladus orientalis* were clearly divided into three groups.

The essential oil content and volatile components of ancient *Platycladus orientalis* leaves of different ages were analyzed. The results showed that the extraction yield of essential oil increased first and then decreased with the increase of tree age, and the extraction yield of essential oil in ancient *Platycladus orientalis* leaves with different tree ages was significantly different. A total of 82 volatile components were identified in ancient *Platycladus orientalis* leaves with different tree ages, with 14 common components. The main components were terpenes. With the increase of tree ages, the ester and alcohol content increased gradually, while the difference between different tree age segments was not significant. The tree age of *Platycladus orientalis* leaves had an important influence on the flavor, content, composition and function of its essential oil. Therefore, under the development and utilization of essential oil in ancient *Platycladus orientalis* leaves, the appropriate processing technology and product type should be selected based on the variation trend of the volatile component composition, and differentiated development and utilization should be carried out.

## 4. Materials and Methods

### 4.1. Plant Materials

Combined with the particularity, rarity, preciousness and right attribute of ancient tree materials, this study comprehensively considered the actual distribution of ancient *Platycladus orientalis* and minimized the damage caused by sample collection. There were 19 ancient *Platycladus orientalis* trees in Xuanyuan Temple (109.28 E, 35.588 N, 865 m) of Huangdi Mausoleum in Yan’an City, Shaanxi Province (Figure 5), among which the ‘HDL1’ was ‘Huangdishouzhibai’ (Figure 6A), ‘HDL2’ was ‘Hanwudiguajiabai’ (Figure 6B), and the others trees are numbered from HDL3 to HDL19. The tree ages were distributed in 2000, 2500, 3000 and 5000 years. In August 2022, healthy young scale leaves in four directions in the middle of the crown of 19 ancient *Platycladus orientalis* were collected, mixed well, numbered and quickly placed in a plastic bag containing discolored silica gel for drying and preservation. The specific number information of the sampled samples is described in Table 2.

### 4.2. Essential Oil Extraction

The ancient *Platycladus orientalis* leaves with different tree ages were ground (30 Hz, 1.5 min) to powder by a grinding machine (MM 400, Retsch). One hundred g of powder was weighed and placed in a round-bottom flask. 700 mL of 0.2% NaCl solution was added, soaked for 1 h, and then distilled by electric heating for 3 h. Heating was stopped after a period of time until the upper essential oil to clarify the reading; the upper essential oil was collected, anhydrous Na_2_SO_4_ was used to absorb excess water, 3 replicates were set up, and its essential oil yield was recorded (Essential oil yield (mL/100 mg) = essential oil volume/powder dry weight.). The essential oil was stored at 4 °C in a refrigerator.

### 4.3. HS–SPME–GC–O–MS Analysis 

The volatile compounds in ancient *Platycladus orientalis* leaves were analyzed and identified using the HS–SPME–GC–O–MS method. 

HS-SPME conditions: each sample was accurately weighed at 1.0 g, placed in a 10 mL headspace bottle, equilibrated at 80 °C for 30 min, and extracted by solid phase micro-extraction needle (100 μL PDMS fiber, SUPELCO, Bellefonte, PA, USA). After the extraction, the fiber was desorbed at the injection port for 5 min for HS–SPME–GC–O–MS analysis.

Gas chromatography (GC) conditions: HP-5MS (30 m × 0.25 mm × 0.25 μm, Agilent, 6890 N-5975B) chromatographic column was used, the injector temperature was 230 °C, the carrier gas was high purity He, the flow rate was 1.0 mL/min, the detector temperature was 280 °C, and the split ratio was 100:1. Temperature program: the inlet temperature was 280 °C, the column temperature was increased from 50 °C to 250 °C at a rate of 2 °C/min, and maintained for 20 min, the solvent delay was 3 min.

Mass spectrometer (MS) conditions were: ionization mode: EI, electron energy 70 eV, the ion source temperature 230 °C, quadrupole temperature 150 °C, interface temperature 230 °C, emission current 34.6 A, acquisition mode: full scan, mass scanning range *m*/*z*, 50–550 u.

The collected mass spectra were searched using the NIST library (NIST Mass Spectral Database 2.2) to identify the volatile components in the samples, and the relative contents of each component was analyzed by the area normalization method.

GC–O identification method: The characteristic aroma components in ancient *Platycladus orientalis* leaves with different tree ages were analyzed by olfactometer and gas chromatography. The GC–O experiment was conducted by three experienced sensory evaluators. During the experiment, at least two sensory evaluators could obtain the same sensory description at the same smelling time, and the record was recorded into the final result.

### 4.4. Data Analysis

Chemometric analyses such as hierarchical cluster analysis (HCA), principal component analysis (PCA) and orthogonal partial least squares discriminant analysis (OPLS–DA) were performed to systematically analyze the influence of tree age on the volatile component contents in ancient *Platycladus orientalis* leaves. Heatmap analysis and principal component analysis were generated using the Origin software for statistical and computing (Origin Pro 2020b, Origin Lab, USA). Orthogonal partial least squares discriminant analysis was performed using R (http://www.r-project.org/ (accessed on 15 November 2022)). Variance analysis was performed using SPSS 24.0 for Windows (SPSS Inc., Chicago, IL, USA).

## Figures and Tables

**Figure 1 molecules-28-02043-f001:**
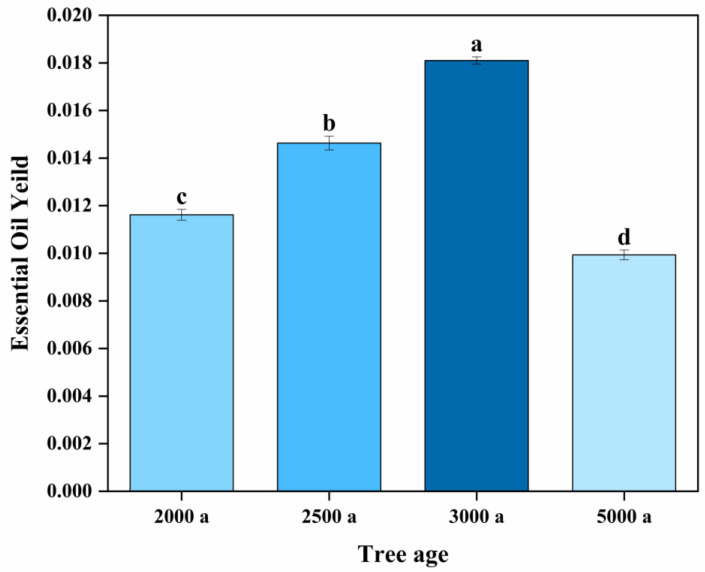
Essential oil yield of ancient *Platycladus orientalis* leaves with different tree ages. Note: Different lowercase letters represented significant differences at the 0.05 level (*p* < 0.05).

**Figure 2 molecules-28-02043-f002:**
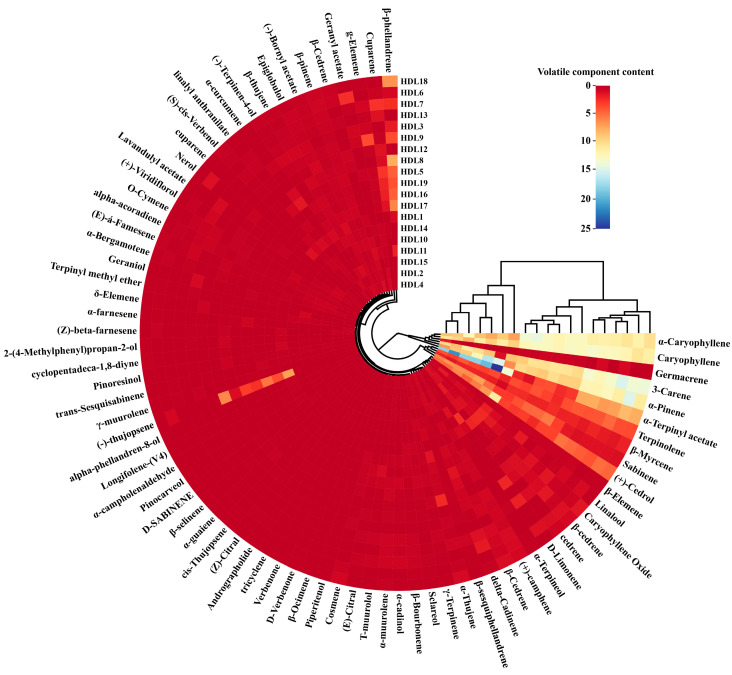
Volatile components content of 19 ancient *Platycladus orientalis* leaves with different tree ages.

**Figure 3 molecules-28-02043-f003:**
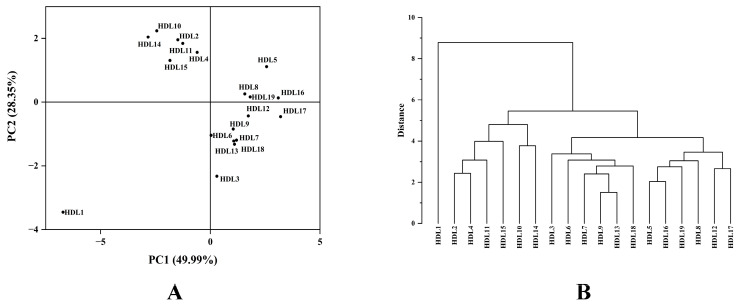
(**A**): 2D-PCA plot and (**B**): HCA plot of 19 ancient *Platycladus orientalis* leaves with different tree ages based on volatile components content.

**Figure 4 molecules-28-02043-f004:**
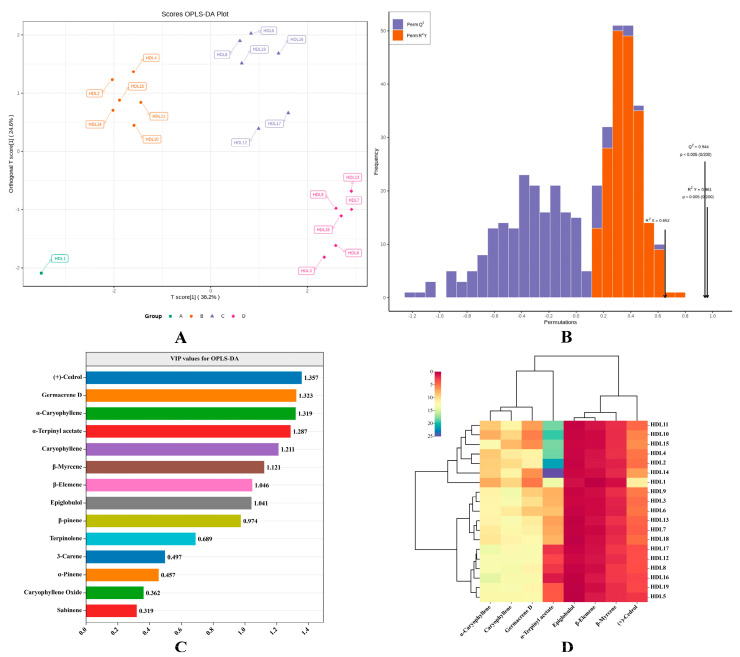
(**A**): Scatter plots and (**B**): 200 permutation tests for OPLS–DA analysis of 19 ancient *Platycladus orientalis* leaves with different tree ages based on volatile components content. (**C**): VIP values of amides compounds for OPLS–DA analysis and (**D**): cluster heat map based on key differential volatile compounds.

**Figure 5 molecules-28-02043-f005:**
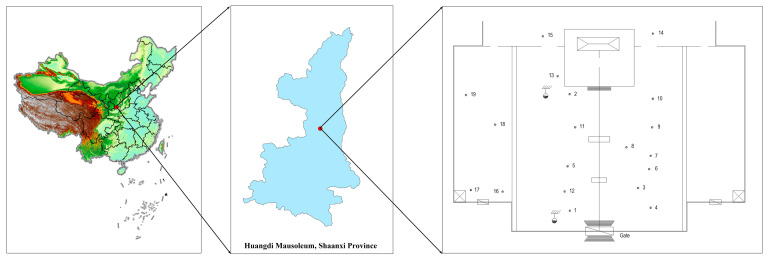
Map of the collection sites of ancient *Platycladus orientalis* in Huangdi Mausoleum.

**Figure 6 molecules-28-02043-f006:**
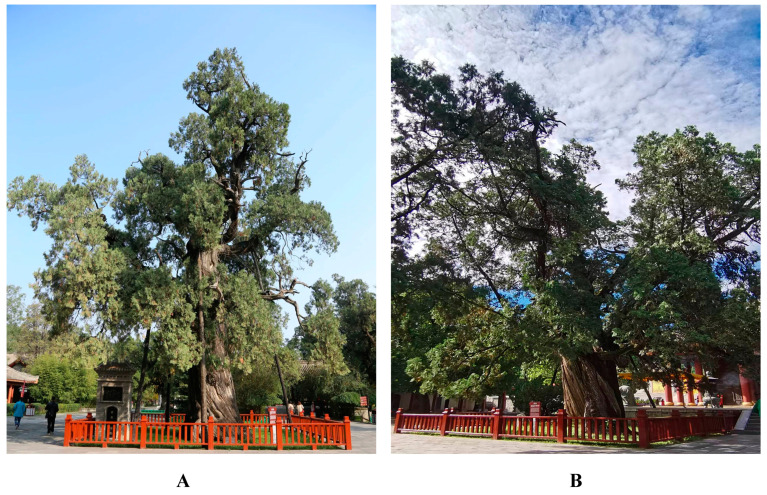
(**A**): ‘HDL1’-‘Huangdishouzhibai’, (**B**): ‘HDL2’-‘Hanwudiguajiabai’.

**Table 1 molecules-28-02043-t001:** The relative content of common volatile components in 19 ancient *Platycladus orientalis* leaves with different tree ages.

Samples	α-Pi	Sa	Ca	Ter	Cary	α-Cary	Germ	Ele	Myr	β-Pi	Cary Oxide	Cedr	Epi	α-Ter
HDL1	6.40	3.02	8.90	2.17	9.61	7.34	5.27	0.17	0.90	0.23	2.28	11.30	0.98	13.88
HDL2	9.02	6.69	8.70	3.25	9.90	9.15	11.40	1.14	1.57	0.32	0.39	4.92	0.44	21.42
HDL4	6.88	5.97	10.17	4.12	10.93	8.81	11.53	1.11	2.13	0.34	0.27	5.17	0.56	18.29
HDL10	16.76	7.29	8.92	3.21	9.55	7.54	5.57	0.75	2	0.57	0.63	5.77	0.54	19.42
HDL11	12.48	6.36	10.41	3.35	11.52	8.81	6.94	1.04	2.05	0.24	0.51	4.57	0.39	18.18
HDL14	14.98	5.04	1.14	3.74	11.47	9.23	6.36	0.86	1.82	0.43	0.69	6.89	0.6	25.68
HDL15	9.43	6.61	11.51	4.25	8.04	12.86	6.17	0.80	2.05	0.37	0.35	6.17	0.64	18.12
HDL3	12.20	1.16	12.56	4.09	12.99	11.91	10.18	0.78	1.62	0.29	1.50	4.88	0.62	7.82
HDL6	14.92	1.24	13.86	3.83	10.88	12.04	8.75	0.75	1.44	0.58	0.63	5.12	0.46	8.42
HDL7	11.56	1.29	15.12	4.95	12.34	10.89	11.46	0.64	1.69	0.49	0.27	4.09	0.37	7.27
HDL9	12.36	1.54	13.67	4.28	13.53	11.64	9.03	0.87	2.07	0.49	0.51	5.36	0.48	7.69
HDL13	12.89	1.31	13.41	4.17	12.91	11.69	11.02	0.95	2.16	0.44	0.47	4.42	0.29	7.01
HDL18	11.4	1.11	14.03	3.73	11.74	10.59	12.45	0.87	2.02	0.49	1.42	4.83	0.23	7.78
HDL5	10.32	4.91	10.99	3.35	12.34	12.29	12.62	1.23	2.07	0.55	0.33	2.34	0.19	3.93
HDL8	9.49	5.68	10.80	2.31	13.18	13.00	11.84	0.88	2.52	0.46	0.61	3.38	0.47	2.47
HDL12	9.53	3.19	10.20	3.87	12.93	13	12.82	1.08	1.99	0.37	1.32	3.36	0.79	2.93
HDL16	9.62	3.98	10.68	3.26	13.1	14.41	13.13	1.22	2.42	0.5	0.98	2.94	0.21	1.29
HDL17	10.23	3.02	9.33	3.84	12.55	13.79	12.97	0.96	2.44	0.55	1.63	3.34	0.5	2.24
HDL19	9.17	4.89	10.55	4.07	12.92	13.12	12.11	0.86	2.05	0.37	0.38	3.14	0.25	3.84
Mean	11.03	3.91	10.79	3.68	11.71	11.16	10.09	0.89	1.95	0.43	0.80	4.84	0.47	10.40
CV/%	23.63%	54.70%	27.33%	17.76%	12.72%	18.50%	26.57%	26.28%	19.10%	24.95%	69.02%	39.41%	41.79%	70.77%

Note: α-Pi—α-Pinene, Sa—Sabinene, Ca—3-Carene, Ter—Terpinolene, Cary—Caryophyllene, α-Cary—α-Caryophyllene, Germ—Germacrene D, Ele—β-Elemene, Myr—β-Myrcene, β-Pi—β-Pinene, Cary Oxide—Caryophyllene Oxide, Cedr—(+)-Cedrol, Epi—Epiglobulol, α-Ter—α-Terpinyl acetate.

**Table 2 molecules-28-02043-t002:** Detailed information on the ancient *Platycladus orientalis* in the Xuanyuan temple of the Huangdi Mausoleum.

Samples No.	Diameter at Breast Height (cm)	Height (m)	Crown Width (m)		Tree Age (a)
			East-West Crown	South-North Crown	
HDL1	760	19.3	14.4	15.3	5000
HDL2	518	14.8	12.6	12.7	3000
HDL3	334	14.0	10.1	13.5	2500
HDL4	510	19.0	9.8	11.2	3000
HDL5	346	18.5	9.3	10.9	2000
HDL6	470	20.5	12.1	15.8	2500
HDL7	502	17.0	13.9	11.8	2500
HDL8	368	18.0	10.9	9.9	2000
HDL9	500	17.0	18.5	11.8	2500
HDL10	516	18.0	12.9	18.5	3000
HDL11	470	21.5	20.2	19.0	3000
HDL12	440	9.5	8.8	13.8	2500
HDL13	579	15.0	16.8	10.9	2000
HDL14	430	11.4	5.4	12.5	3000
HDL15	500	12.8	10.1	12.3	3000
HDL16	525	12.0	14.6	9.2	2000
HDL17	388	16.0	13.1	13.2	2000
HDL18	430	15.0	11.7	9.8	2500
HDL19	540	20.0	15.0	10.0	2000

## Data Availability

The data used to support the findings of this study are available from the corresponding author upon reasonable request.

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
