# Peer review of "Chemotaxonomic Variation in Volatile Component Contents in Ancient Platycladus orientalis Leaves with Different Tree Ages in Huangdi Mausoleum"

_molecules, 2023, doi:10.3390/molecules28052043_

Round 1

Reviewer 1 Report

In the present study, the essential oil content of old leaves of Platycladus orientalis was extracted by steam distillation, and the composition of the volatile components of the leaves was analyzed by solid-phase microextraction combined with gas chromatography-mass spectrometry (HS-SPME-GC-MS).

The main goal of the work is to study the composition of the extracts to try to find significant differences that allow differentiating these trees according to their age. Of the 82 compounds identified, the chemometric treatment allows to find the 8 significant variables that allow this differentiation.

he chemometric studies as well as the results obtained and their discussion are well presented and, from my point of view, the work can be published in the journal after minor revision.

Comment

1.- Figure 1 is not well explained, more information about it is needed for the reader to understand it properly. For example, what do the letters "a", "b", "c" and "d" mean? In what units the essential oil yield is expressed? There is no correspondence with other data appearing in the paper. On the other hand, how is the sensory analysis carried out and is there a panel of experts? The information in this part of the paper should be completed.

Minor comments

2.- Figure 3C is not identified in the text (section 2.5).

3.- Table 1 should be presented in a better format, so that it is more visual and easier to read. 

4.- From my point of view, tables S2, S3 and S4 in the supplementary material folder are unnecessary. All the important information is contained in table S1. The text should be written according to the elimination of this material.

Author Response

1.- Figure 1 is not well explained, more information about it is needed for the reader to understand it properly. For example, what do the letters "a", "b", "c" and "d" mean? In what units the essential oil yield is expressed? There is no correspondence with other data appearing in the paper. On the other hand, how is the sensory analysis carried out and is there a panel of experts? The information in this part of the paper should be completed.

Response: Note:Different lowercase letters represented significant difference at 0.05 level (P<0.05). Essential oil yield rate (ml/100mg) = essential oil volume/ powder dry weight. The content composition of essential oil in ancient Platycladus orientalis leaves with different tree ages was significantly different. Therefore, we decided to use GC-MS analysis to qualitatively and quantitatively detect the essential oil components, which was helpful to develop products with exquisite efficacy and strong functionality. GC-O identification method: The characteristic aroma components of samples were analyzed by olfactometer and gas chromatography. The split ratio between mass spectrometry and sniffers was 1:1. GC-O analysis was performed by three experienced sensory evaluators. In the analysis process, at least two sensory evaluators can obtain the same sensory description at the same olfactory time, and then the final results of the record are recorded.

Figure 3C is not identified in the text (section 2.5).

Response: The VIP values could quantify the contribution of each variable factor to the classifica-tion. The larger the VIP value, the more significant the contribution of volatile compo-nents to the difference of ancient Platycladus orientalis leaves samples with different tree ages. Though calculating the VIP values, it was found that VIP values of 8 com-mon volatile components were greater than 1 (Figure 4C).

Table 1 should be presented in a better format, so that it is more visual and easier to read.

Response: Table 1 was to illustrate the common volatile components content in ancient Platycladus orientalis with different tree ages, and the coefficient of variation of different components. Names superior to volatile components were too long, so abbreviations were used instead.

From my point of view, tables S2, S3 and S4 in the supplementary material folder are unnecessary. All the important information is contained in table S1. The text should be written according to the elimination of this material.

Response: Table S1 was to reveal the volatile components in ancient Platycladus orientalis leaves with different tree ages from a holistic perspective. Table S2, Table S3 and Table S4 were specifically to explain the composition of terpenes, alcohols and esters, respectively.

Reviewer 2 Report

In this manuscript, the authors try to reveal the relationship between the volatile compound composition and tree ages using gas chromatography-mass spectrometry incorporated with the SPME method. OPLS-DA analysis and HCA method were conducted based on the profile of identified volatile chemicals. The experimental design is straightforward and reasonable, and the conclusions are mainly based on the data analysis.

1 In the manuscript, such as L100 and L 352, the expression ‘extraction rate’ may be confused with ‘extraction yield’ based on their description of their experiments. What is the unit for the ‘extraction rate’?

2 From Fig.1, the yields of essential oil are above 0.01 for all tested samples, why not the authors just use the essential oil for GC-MS analysis?

3 The authors used just ONE SPME needle for the analysis, and the chemical information may not be comprehensive enough to compare the differences between the tested samples. Also, the authors should clarify ‘volatile components’ and ‘essential oil’ in the whole manuscript.

4 The general description of OPLS-DA and HCA in L293 to L306 are not directly related to the manuscript and could be abridged.

5 The authors only conducted a limited experiment about the volatile compounds using SPME-GC-MS, and the essential oil is not fully characterized. Thus, the conclusion comment about the utilization value of essential oil should be very careful.

Author Response

1 In the manuscript, such as L100 and L 352, the expression ‘extraction rate’ may be confused with ‘extraction yield’ based on their description of their experiments. What is the unit for the ‘extraction rate’?

Response: In the manuscript, we have used extraction yield instead of extraction rate. And, Essential oil yield (ml/100mg) = essential oil volume/ powder dry weight, the unit for the essential oil yield was ‘ml/100mg’.

Line 43: From Fig.1, the yields of essential oil are above 0.01 for all tested samples, why not the authors just use the essential oil for GC-MS analysis?

Response: Headspace (solid-phase microextraction) technology is mainly used for the detection of volatile or semi-volatile components, odorous substances and aroma substances in gas, liquid or solid samples. Therefore, dry powder was applied to determine the volatile components in ancient Platycladus orientalis leaves with different tree ages.

Line 50: The authors used just ONE SPME needle for the analysis, and the chemical information may not be comprehensive enough to compare the differences between the tested samples. Also, the authors should clarify ‘volatile components’ and ‘essential oil’ in the whole manuscript.

Response: SPME fibers could be directly inserted into the liquid sample or stay above the sample for headspace sampling. In headspace sampling, the SPME fiber is equivalent to a ‘chemical pump’, which inhaled the compound from the liquid phase into the headspace and then into the fiber. In addition, the measured data diaplayed that there were abundant differences in the composition of volatile components in ancient Platycladus orientalis leaves with different tree ages. Volatile components are substances identified by HS-SPME-GC-MS method in ancient Platycladus orientalis leaves with different tree ages. For example, a total of 72 volatile components were isolated and identified in 19 ancient Platycladus orientalis leaves with different tree ages, and 14 common volatile components were screened. The essential oil is extracted to analyze the yield of essential oil from ancient Platycladus orientalis leaves with different tree ages. For example, the essential oil extracted by steam distillation was pale yellow, and the yield of essential oil increased first and then decreased with the increase of tree age. In response to this problem, we have made changes in the manuscript.

The general description of OPLS-DA and HCA in L293 to L306 are not directly related to the manuscript and could be abridged

Response: In this study, OPLS-DA and HCA statistical methods were applied to evaluate the differences in volatile components in 19 ancient Platycladus orientalis leaves with different tree ages, and to screen and distinguish the differential volatile substances in 19 ancient Platycladus orientalis leaves with different tree ages. We considered that it was reasonable to appear in the discussion.

The authors only conducted a limited experiment about the volatile compounds using SPME-GC-MS, and the essential oil is not fully characterized. Thus, the conclusion comment about the utilization value of essential oil should be very careful.

Response: We have modified the conclusion to “Therefore, under the development and utilization of essential oil in ancient Platycladus orientalis leaves, the appropriate processing technology and product type should be selected based on the variation trend of the volatile component composition, and differentiated development and utilization should be carried out”.

Reviewer 3 Report

The authors investigated the volatiles in ancient Platycladus orientalis leaves using HS-SPME-GCMS, and explored the discrimination in different tree ages. The study is of interest for readers. Before acceptance for publication in Molecules, some minor revisons should to be clarified.

1. Why HS-SPME was selected to be used for detecting the volatiles in this study? 

2. The HS-SPME conditions e.g. extraction temperature, time, should to be clarified in the experimental section.

3. Some references about the SPME for volatiles analysis should be concluded, e.g. DOI: 10.1016/j.microc.2021.106133 , DOI: 10.1016/j.aca.2022.340159,  DOI: 10.1016/j.foodchem.2012.04.053.

Author Response

  1. Why HS-SPME was selected to be used for detecting the volatiles in this study?

Response: Headspace (solid-phase microextraction) technology is mainly used for the detection of volatile or semi-volatile components, odorous substances and aroma substances in gas, liquid or solid samples. For some volatile components and semi-volatile components sensitivity is better, can get stable and reliable qualitative and quantitative results. For some low-content, low-volatile components, solid-phase microextraction technology can effectively play an enrichment role. Therefore, HS-SPME-GC-MS was applied to determine volatile components in ancient Platycladus orientalis leaves with different tree ages.

  1. The HS-SPME conditions e.g. extraction temperature, time, should to be clarified in the experimental section.

Response: Each sample was accurately weighed at 1.0 g, placed in 10 mL headspace bottle, equilibrated at 80°C for 30 min, and extracted by solid phase micro-extraction needle (100 μL PDMS fiber, SUPELCO, USA). After the extraction, the fiber was desorbed at the injection port for 5 min for HS-SPME-GC-O-MS analysis.

  1. Some references about the SPME for volatiles analysis should be concluded, e.g. DOI: 10.1016/j.microc.2021.106133,DOI:10.1016/j.aca.2022.340159,DOI:10.1016/j.foodchem.2012.04.053.

Response: We have added the above references in the manuscript.
